# Novel Microdialysis Technique Reveals a Dramatic Shift in Metabolite Secretion during the Early Stages of the Interaction between the Ectomycorrhizal Fungus *Pisolithus microcarpus* and Its Host *Eucalyptus grandis*

**DOI:** 10.3390/microorganisms9091817

**Published:** 2021-08-26

**Authors:** Krista L. Plett, Scott Buckley, Jonathan M. Plett, Ian C. Anderson, Judith Lundberg-Felten, Sandra Jämtgård

**Affiliations:** 1Hawkesbury Institute for the Environment, Western Sydney University, Richmond, Hawkesbury, NSW 2753, Australia; j.plett@westernsydney.edu.au (J.M.P.); i.anderson@westernsydney.edu.au (I.C.A.); 2Elizabeth Macarthur Agricultural Institute, New South Wales Department of Primary Industries, Menangle, Wollondilly, NSW 2568, Australia; 3Department of Forest Ecology and Management, Swedish University of Agricultural Sciences, SE-901 83 Umeå, Sweden; scott.buckley@slu.se (S.B.); sandra.jamtgard@slu.se (S.J.); 4Umeå Plant Science Centre, Department of Forest Genetics and Plant Physiology, Swedish University of Agricultural Sciences, SE-901 83 Umeå, Sweden; judith.lundberg-felten@slu.se

**Keywords:** metabolomics, pre-symbiosis signalling, ectomycorrhizal fungi, microdialysis

## Abstract

The colonisation of tree roots by ectomycorrhizal (ECM) fungi is the result of numerous signalling exchanges between organisms, many of which occur before physical contact. However, information is lacking about these exchanges and the compounds that are secreted by each organism before contact. This is in part due to a lack of low disturbance sampling methods with sufficient temporal and spatial resolution to capture these exchanges. Using a novel in situ microdialysis approach, we sampled metabolites released from *Eucalyptus grandis* and *Pisolithus microcarpus* independently and during indirect contact over a 48-h time-course using UPLC-MS. A total of 560 and 1530 molecular features (MFs; ESI- and ESI+ respectively) were identified with significant differential abundance from control treatments. We observed that indirect contact between organisms altered the secretion of MFs to produce a distinct metabolomic profile compared to either organism independently. Many of these MFs were produced within the first hour of contact and included several phenylpropanoids, fatty acids and organic acids. These findings show that the secreted metabolome, particularly of the ECM fungus, can rapidly shift during the early stages of pre-symbiotic contact and highlight the importance of observing these early interactions in greater detail. We present microdialysis as a useful tool for examining plant–fungal signalling with high temporal resolution and with minimal experimental disturbance.

## 1. Introduction

Trees in forest ecosystems worldwide live in symbiosis with ectomycorrhizal (ECM) fungi, which can assist the tree in the acquisition of limiting nutrients and trace elements [1,2,3]. The formation of a mature colonised root system is the culmination of many signalling exchanges between the plant host and the ECM fungus [4,5]; complex transcriptomic reprogramming [6,7] as both tree root and fungus are structurally altered to form the new hybrid symbiotic root tissue. Many of these signalling exchanges and transcriptomic alterations occur even before the two organisms come into physical contact [8,9,10], and some of these signals have been reported to be of volatile nature [11]. Interruptions, or “miscommunication” at these early stages of interaction, may impact the final outcome of the symbiosis and may explain why certain fungal-host combinations are incompatible [12,13]. 

Despite the importance of these early signalling events, there are still many gaps in our understanding of the various signals passed between the interacting organisms. Tree roots secrete sugars and other carbon-rich compounds into the rhizosphere, which attract and are used not only by ECM fungi but also other microorganisms in the soil [14,15]. Other metabolites, such as flavonoids, have roles in promoting symbiosis by encouraging ECM spore germination [16] or the production of symbiosis specific proteins in the ECM fungus [17]. ECM fungi release molecules including sesquiterpenes [11] and phytohormones such as auxin [18,19] that increase lateral root developmentduring pre-contact. Recently, lipochitooligosaccharides have been demonstrated to be released by the ECM fungus L. bicolor, and the host root’s capacity to recognise and respond to these molecules indicates that these are another important signalling route during ECM formation [20]. In addition to metabolites, fungal proteins, such as effector proteins, are also known to be transferred to plant tissues where they can alter gene transcription or plant function [21,22,23,24,25]. However, these metabolites and proteins are just a few of the many signals passed between organisms. Recent work demonstrated that within 24 h of indirect contact between ECM fungi and tree roots, hundreds of fungal-produced metabolites are already found within the tree root tissues and, further, that the quantity and identity of these metabolites co-vary with the colonisation success of the ECM fungus [8,26]. The vast majority of these remain uncharacterised. 

One of the factors complicating the analysis of metabolites involved in the interaction between host trees and ECM fungi is the difficulty in sampling metabolites that may be present only transiently or in very low concentrations [27]. Previous studies have collected root and fungal exudates in liquid media or hydroponic solutions (e.g., [28,29,30]) or have used solid-phase extraction techniques to sample volatile metabolites [31]. Alternatively, metabolites can be extracted from the plant root tissues directly [8,13,26]. Here, we explore the use of microdialysis as a technique to sample metabolites directly from the plant–fungal interface with little perturbation to the system and in real-time. Originally designed for use in neuroscience, microdialysis probes have recently been used in more ecological applications for assessing soil nitrogen, metabolites and enzymes [32,33,34]. This technique allows for continuous in situ samplings of soluble compounds by induced diffusion across the probe’s semi-permeable membrane. The probe is similar in scale to a plant root with a diameter of 0.5 mm and has the advantage of being able to sample metabolites directly from the zone of interest without greatly affecting the experimental system. 

Using microdialysis, we have captured metabolites secreted during the interaction of *Eucalyptus grandis* with the ECM fungus *Pisolithus microcarpus* over a time-course for the first 48-h of pre-symbiotic contact. With the availability of a sequenced genome for both organisms and the ability to form mycorrhizal root tips readily in vitro, *E. grandis* and *P. microcarpus* are important model species for the molecular study of ECM interactions [35,36,37,38,39]. Our results highlight not only the usefulness of microdialysis as a method for sampling metabolites but also the rapid and dramatic shift in the secreted metabolome of ECM fungi upon detection of a potential plant host.

## 2. Materials and Methods

### 2.1. Biological Materials and Growth Conditions

*Eucalyptus grandis* seeds (CSIRO tree seed centre, lot 21068) were sterilised in 30% hydrogen peroxide for 10 min and rinsed three times in sterile water prior to plating on 1% agar media in a biological safety cabinet. The seeds were placed in a walk-in growth chamber (Kryo-Service Oy, Helsinki, Finland) for four weeks at 12-h day/night cycles (25 °C day/22 °C night temperature, 70% humidity, light intensity 250 µmol/m^2^/s). Seedlings were then transferred to new 120 mm square Petri plates containing ½ strength modified Melin–Norkrans media (MMN; 0.25 g/L (NH_4_)_2_HPO_4_, 0.15 g/L KH_2_PO_4_, 0.07 g/L of MgSO_4_.7H_2_O and 1 g/L glucose combined with 0.5 mL/L of CaCl_2_ 5% stock solution, 0.5 mL/L of NaCl 2.5% stock solution, 0.5 mL/L of ZnSO_4_ 0.3% stock solution, 66 μL/L of thiamine 0.1% stock solution and 0.5 mL/L of citric acid + FeEDTA 1.25% stock solution brought to pH 5.5 before the addition of 13 g/L of agar). Each plate contained three seedlings, with sterile cellophane membranes placed above and below the seedling roots to prevent drying out and growth of roots into the media. Seedlings were allowed to grow for an additional four weeks in the same chamber conditions as above.

*P. microcarpus* isolate Si14 [39] was sub-cultured onto full strength MMN (0.50 g/L (NH_4_)_2_HPO_4_, 0.30 g/L KH_2_PO_4_, 0.14 g/L of MgSO_4_.7H_2_O and 10 g/L glucose combined with 1 mL/L of CaCl_2_ 5% stock solution, 1 mL/L of NaCl 2.5% stock solution, 1 mL/L of ZnSO_4_ 0.3% stock solution, 133 μL/L of thiamine 0.1% stock solution and 1 mL/L of citric acid + FeEDTA 1.25% stock solution brought to pH 5.5 before the addition of 13 g/L of agar) and allowed to grow at room temperature for one month. Small squares (1 cm × 1 cm) were excised from the leading edges of the growing colonies and placed onto fresh 90 mm round Petri plates containing ½ strength MMN media (1 g/L glucose) covered with a sterile cellophane membrane. Plates were allowed to grow for two weeks at 28 °C. 

### 2.2. Experimental Design

Metabolites secreted by roots or fungal colonies during the early stages of pre-symbiotic contact were monitored using a microdialysis probe (30 mm × 0.5 mm, 20 kDa MWCO, CMA20, CMA Microdialysis AB, Solna, Sweden) placed next to plant roots and/or the fungal colonies in a petri-plate-based microcosm. Pre-symbiotic indirect contact plates containing both fungal colonies and plants (10 biological replicates), fungal-only plates (3 biological replicates), plant-only plates (6 biological replicates) and media-only control plates were assembled with one microdialysis probe per plate to capture metabolites secreted by the organism(s). All experiments were done in 120 mm square Petri plates containing ½ strength MMN with low glucose (0.1 g/L). Holes were burned in the upper corner of each Petri plate to allow for the unrestricted exit of the microdialysis probe leads. 

For the pre-symbiotic indirect contact plates, the fungal colonies and plant roots were layered with a semi-permeable cellophane membrane between them, which allowed the passage of signalling molecules and small proteins but prevented physical contact between the two organisms, mimicking the pre-symbiotic phase. The plates contained one cellophane membrane on top of the media, followed by an 8-week-old E. grandis seedling, the microdialysis probe, another layer of cellophane and then the two-week-old fungal colony (Figure 1A). Fungal-only plates contained a cellophane membrane, followed by the microdialysis probe and an additional cellophane membrane with the growing fungal colony. Plant-only plates contained a cellophane membrane, the plant and microdialysis probe and a second membrane, and, finally, the media-only control plates contained a cellophane membrane, the microdialysis probe and an additional cellophane membrane. Approximately 50 μL of sterile MilliQ water was added to each plate around the microdialysis probe to ensure good contact between cellophane membrane layers and the probe and encourage the free movement of metabolites.

All plates were returned to the growth chamber, and the microdialysis probes were connected to infusion pumps (CMA 4004; CMA Microdialysis AB, Solna, Sweden) pumping sterile MilliQ water through the probes at a rate of 5 µL min^−1^ (Appendix A). The resulting solution from each probe containing the metabolites was collected in a 2 mL screw cap vial kept on ice. Sampling occurred over an hour to collect sufficient samples, and afterwards, samples were frozen at −80 °C. Microcosms were all sampled at five points of time, 0 h, 8 h, 24 h, 32 h and 48 h. The experiment began in the morning such that all measurements occurred during the daylight hours of the plant’s circadian cycle. 

### 2.3. Metabolite Analysis

Samples collected from the microdialysis probes for metabolite analysis (200 µL of each sample) were freeze-dried for 48 h (LaboGene CoolSafe Pro 110-4, Allerød, Denmark). After freeze-drying, samples were stored at −80 °C until analysis. Dried samples were resuspended in 10 μL of cold water/methanol/formic acid solution (50/49.9/0.1), and each sample was vortexed and sonicated for five minutes. This solution was directly injected onto a Waters nanoACQUITY UltraPerformance Liquid Chromatography system (Waters, Wilmslow, UK) coupled to an SYNAPT G2-S Mass Spectrometer (Waters, Wilmslow, UK). Samples were run alongside a pooled biological quality control sample (created by combining small aliquots from several samples across all treatment types), method blanks (from media only microcosms) and true blanks (resuspension solution only), running the instrument in both negative (ESI-) and positive (ESI+) electrospray ionisation modes. The instrument was operated in high-resolution mode integrated with ion mobility to enhance the separation of ions.

All data pre-processing, including peak alignment, peak picking and deconvolution, was conducted with the Progenesis QI programme (Nonlinear Dynamics Ltd., Newcastle upon Tyne, UK). Pre-filtering of the dataset was performed to remove background, low abundance features or artefacts by selecting features with a fold change of at least 10 between any of the four treatments (indirect contact, plant-only, fungal-only or media-only control), *p*-values < 0.05 (ANOVA) and excluding features that had the highest average abundance in the media blanks. Data transformation (log-transformation and Pareto scaling) and multivariate analyses (PCA, OPLS-DA; [40]) were performed using SIMCA 16 (Sartorius Stedim Data Analytics AB, Umeå, Sweden). Variable-in-projection scores > 1 were used to determine significant features in OPLS-DA models for each ionisation mode. Univariate analyses (ANOVA; repeated measures ANOVA) were performed using GraphPad Prism 8.4.2 (GraphPad Software LLC., San Diego, CA, USA, 2020); Dunnett’s Post Hoc tests were used to differentiate significant features (*p* < 0.05) in samples with plant/fungus or both from those features also detected in control group background; Tukey’s Post Hoc tests were used to identify significant features (*p* < 0.05) in the pre-symbiotic indirect contact condition from those in plant- and fungus-only conditions. Heat maps were produced either with Metaboanalyst (Figure 2; www.metaboanalyst.ca, accessed on 15 March 2021), with hierarchical clustering of the group averaged by a Euclidean distance measure and using the Ward method for clustering, or using GraphPad Prism 8.4.2 (Figure 4; GraphPad Software LLC., USA), using pre-calculated Z-scores for each feature and sorted by hierarchical clustering (within the indirect contact condition) using Ward distances (SIMCA 16, Sartorius Stedim Data Analytics AB, Umeå, Sweden).

Features were putatively identified using mass and fragmentation scores in Progenesis QI, matching the following databases: HMDB4, MONA, LipidMaps, Metlin, Kegg and CCS (Progenesis library) using a mass error tolerance of 5 ppm, and fragmentation scores > 30. Metabolites of synthetic origin or drug or animal-derived were excluded. Classes were assigned to features based on common chemical structures when multiple matches were presented.

## 3. Results

### 3.1. Indirect Contact between E. grandis and P. microcarpus Alters the Secretion of Secondary Metabolites in Both Organisms with the Induction of a New Set of Metabolites

Microcosms were set up with the two organisms in indirect contact, along with relevant fungal-, plant- and media-only controls (Figure 1A,B) to capture the metabolites secreted and transferred between *E. grandis* seedlings and *P. microcarpus* at the early stages of pre-symbiotic contact. Using untargeted metabolite profiling by UPLC-MS/MS in ESI- and ESI+ mode, a total of 3885 and 3124 molecular features (MFs; ESI- and ESI+ respectively) were identified in the microdialysis samples obtained from the microcosms after 48 h of incubation, with 560 and 1530 of those MFs (ESI- and ESI+ respectively) showing significantly increased differential abundance in at least one test condition when compared to media-only controls (Dunnett’s Post Hoc test, *p* < 0.05; Figure 1C).

Representing the relative abundance of the 560 and 1530 MFs over all conditions in a heatmap (Figure 2A,B), it is evident that the MFs were usually predominant in only a single sample type. The large majority of MFs had a high relative abundance in fungus-only samples, whereas they were low in plant and indirect contacts. A minor proportion of MFs showed high abundance in indirect contacts as compared to plant- and fungus-only samples. Only very few MFs showed higher relative abundance in plant-only samples than in fungus-only or indirect contacts. It is interesting to note that even though a large number of MFs might be shared between fungus-only and indirect contact conditions, they were usually more abundant, specifically in one of the two conditions, especially in ESI- mode, indicating that the fungal secreted metabolome shifts substantially when a plant is present.

To better identify the molecular features that correlated with fungal-only, plant-only or pre-symbiotic indirect contact conditions, we used multivariate analysis approaches with the ESI- and ESI+ data. Principal component analysis (PCA) of the ESI- features showed separation of the metabolic profile obtained from each of the conditions with some outliers in the indirect contact condition (Figure 3A), while for ESI+ MFs, only the fungal control condition separated from the others (Figure 3B). Using a supervised class model (OPLS-DA), there was a clear separation of metabolite profiles between conditions, though with some outliers as with the PCA analysis (Figure 3C,D; Appendix A). Loadings plots derived from the OPLS-DA analysis (Figure 3E,F) show many MFs associated with the fungal-only conditions and a smaller grouping associated with plant-only or indirect contact conditions, depending on the ionisation mode. The MFs most strongly associated with each condition are listed (Appendix A). While the identity of many of these MFs remains unknown, within the fungal-only or indirect contact conditions several lipids, phenylpropanoid or flavonoid compounds were identified, while in the plant-only condition, organic acids and glycerophospholipids were identified.

### 3.2. Secondary Metabolites Induced by Indirect Contact between E. grandis and P. microcarpus Appear within Hours of Contact and Increase in Concentration over Time

Based on the analysis above, we isolated a further 88 MFs that were identified as abundant in the pre-symbiotic indirect contact condition and significantly greater than other conditions at 48-h (ANOVA, Tukey’s Post Hoc test, *p* < 0.05) for further examination of abundance patterns over time (Figure 4). Most of these features were from the ESI- mode data set. In general, the relative abundance of the identified MFs was already elevated over control samples by the 0 or 8-h time point and increased steadily over the 48 h incubation. A few features were found to peak in abundance earlier, around 24-h, or to be most abundant at 0-h and steadily decrease from there (namely an MF in the Glycerophospholipid class (positive mode) and an unidentified MF (negative mode)). These latter MFs may be associated with a disturbance to the organisms during the set-up of the microcosm as similar 0-h maxima were also found to a lesser extent in the plant- or fungal-only sample types. Interestingly, the 32-h timepoint often had reduced metabolite abundances compared to either the 24- or 48-h timepoints. Compound classes or putative identifications were assigned to the MFs as available. While most remain unidentified, among those with classifications assigned to them, there were several phenylpropanoids, organic acids and fatty acids.

## 4. Discussion

Early signalling between ECM fungi and plants is crucial to symbiotic outcomes; however, little is known about the many signalling molecules and metabolites passed between the organisms or the effects they may have. This is partly because of experimental challenges in sampling the metabolites in a manner that is sufficiently sensitive and time-resolved. Here, we explore a new method, microdialysis, to sample metabolites present at the interface between the ECM fungus *P. microcarpus* and its host *E. grandis*. Using this method, we were able to non-destructively sample and detect many metabolites across a 48-h early interaction time-course and demonstrate that the presence of a potential host plant radically alters *P. microcarpus* exudation with several metabolites specifically induced by contact between the two organisms appearing within the first hour of contact. 

A microdialysis-based sampling at the interface between the ECM fungus and its host resulted in the detection of a similar number of MFs compared to other related metabolomics studies sampling ectomycorrhizal root tissues or soil [8,41]. This highlights not only the effectiveness of the microdialysis probe in the sampling of a broad range of metabolites but also the high complexity of the secretome of these two organisms. Considerably fewer molecular features were found in plant-only controls compared to fungal-only controls, potentially because most plant exudates are secreted from plant root tips, only a few of which would have been in contact with the microdialysis probe [15]. Moreover, in this system, the fungus covers a larger area as compared to the plant roots, which may therefore lead to a higher number of fungal metabolites present and detectable. Alternatively, this may simply reflect the quantity and diversity of metabolites secreted by the ECM fungus. Interaction between the two organisms resulted in a dramatic alteration to the metabolites secreted by the fungus, with those metabolites having a high abundance in fungus-only conditions being almost completely replaced with a new set of metabolites. In a metabolomics study with orchid mycorrhizal fungi, Ghirardo et al. [42] similarly found a substantial number of metabolites that were associated with symbiotic tissues but not found in free-living controls, and vice versa. Other studies considering the transcriptomic response of tree hosts to ECM fungi at an early pre-symbiotic time point found a significant enrichment in gene expression associated with alterations to metabolic compound processes in the plant or up-regulation of genes in secretion or general metabolism pathways [8,31,43]. While studies on early plant–microbe interactions focus on the impact of microbe elicitors on plant exudation, our results here demonstrate that the impact of the plant on fungal exudation may be just as important.

The use of an interaction time-course demonstrated that many MFs were induced by pre-symbiotic contact between the two organisms during the first hour of interaction (the 0-h timepoint) compared to plant- and fungus-only conditions. This points to a rapid response to external signalling on the part of the interacting organisms. Other ECM interaction studies have reported significant metabolomic or transcriptomic changes in ECM fungi or hosts within the first 6 to 24 h of interaction [6,8]; however, to our knowledge, responses on the order of minutes have not been previously studied in ECM interactions. In addition to the rapid evolution of metabolites, the 32-h time point was interesting in that, in many cases, metabolites were present in lower abundances than in the 24- or 48-h time points. This may be due to natural diurnal cycling, as the 24- and 48-h time points were taken in the morning, while the 32-h time point occurred in the late afternoon. Light cycling can have large effects on root exudation and, in turn, affect the behaviour and composition of the rhizospheric community [43,44], so it would be interesting if the length of light exposure also affected the metabolites in this experiment. It is also possible that the lower metabolite abundance at 32-h is related to the length of time between samplings. As the previous sample was taken only 8 h before and may have caused a local depletion of metabolites, they may not have built up again to the same extent. If this was the case, it would indicate that many of these metabolites found in the interface between organisms persist over time and gradually increase in concentration. 

Of the metabolites identified with enriched abundance in the indirect contact treatments, several were assigned as phenylpropanoids (i.e., tannin, flavonoids, coumarin, flavylium, phenylpropanoic acid), organic acids or fatty acids (Figure 4). Phenylpropanoids are precursors to a wide range of compounds, including flavonoid compounds, associated with stress and defence responses in plants [45,46]. Plant contact with ECM fungi has previously been shown to induce phenylpropanoid metabolism, and it has been suggested that the presence of these metabolites aids in limiting over-colonisation by the fungus [47,48,49]. However, more recent work involving transgenic down-regulation of genes within the phenylpropanoid pathway suggests that production of these compounds is, in fact, beneficial for mycorrhiza formation, though this could also stem from secondary hormone effects [50]. Flavonoids are also important attractants of ECM fungi, inducing spore germination, hyphal growth or the production of effector proteins within the fungus [16,17,51]. Interestingly, in line with our finding of altered metabolites of the phenylpropanoid pathway, a number of genes related to this pathway were differentially expressed in *E. grandis* roots at 24 h of indirect contact with *P. microcarpus* [8]. Previous metabolomic studies on ECM fungi have also found increased amounts of organic and fatty acids present during plant–fungal interactions, although the role of these compounds in symbiosis is less understood [8,29].

## 5. Conclusion

Overall, the use of microdialysis as a method of detection for metabolites has great potential to be able to answer questions about plant-fungal signalling in ECM and related interactions in a time resolved manner. We have shown that early metabolic signalling, commencing in the first hours of indirect contact, causes substantial shifts in metabolite secretion in the *E. grandis-P. microcarpus* interaction, highlighting the importance of further study into early symbiotic interactions, both within this system and other fungal-host pairings.

## Figures and Tables

**Figure 1 microorganisms-09-01817-f001:**
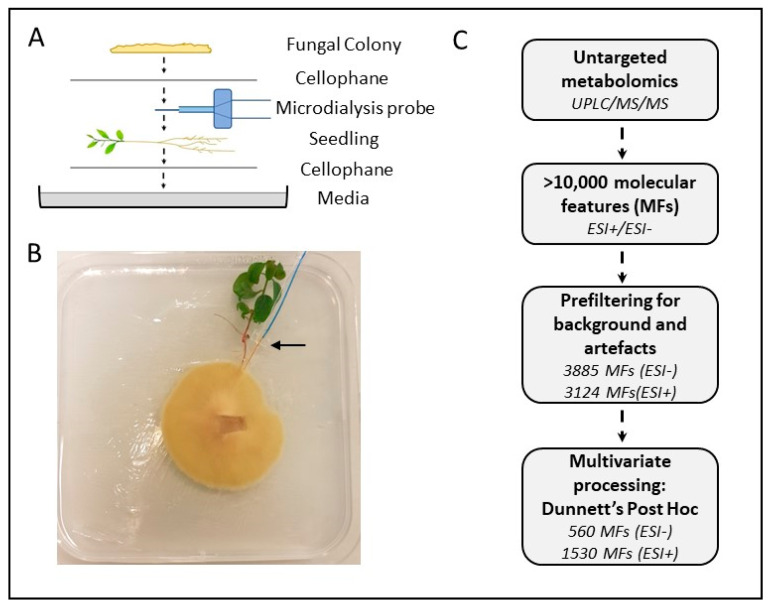
Experimental set-up and data flow for microdialysis-based analysis of metabolites. Schematic (**A**) and photograph (**B**) of experimental set-up for microdialysis probe-based sampling of secreted metabolites from the indirect contact between *P. microcarpus* (fungal colony) and *E. grandis* (seedling). Arrow indicates the location of the microdialysis probe. (**C**) Flow chart describing major steps in analysing and filtering the metabolomics data obtained from the 48-h time point samples for ESI- and ESI+ modes.

**Figure 2 microorganisms-09-01817-f002:**
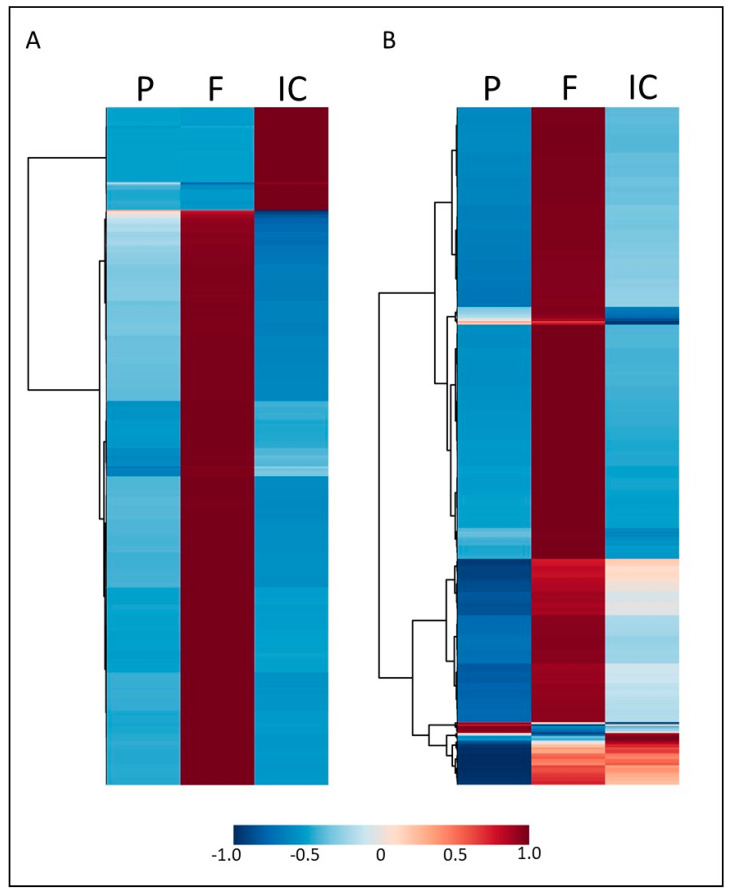
Distribution and abundance of metabolites across sample types. (**A**,**B**) Heatmaps showing the relative abundance of each MF (with hierarchical clustering) across the experimental sample types for ESI- (**A**) and ESI+ (**B**). Sample types are plant-only (P), fungus-only (F) or indirect contact (IC).

**Figure 3 microorganisms-09-01817-f003:**
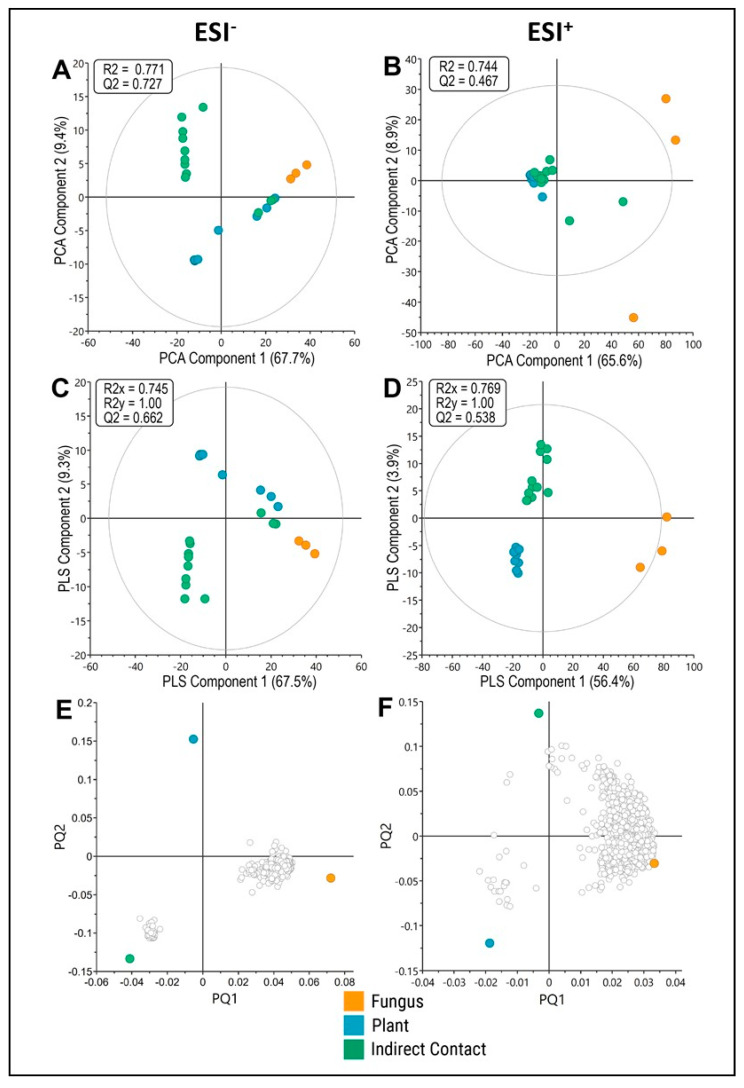
Multivariate analysis of metabolomic data sets. (**A**,**B**) PCA score plots for ESI- (**A**) and ESI+ (**B**) mode data, showing the first two components of metabolite profiles. (**C**,**D**) OPLS-DA score plots for ESI- (**C**) and ESI+ (**D**) mode data, showing the first two components of a supervised classification model and separation of metabolite profiles. (**E**,**F**) Loading plots derived from OPLS-DA of 560 ESI- mode features (**E**) or 1530 ESI+ mode features (**F**). Conditions are indicated in colours: fungus-only (orange), plant-only (blue), and indirect contact (green). Features were sampled at 48 h after start of experiment.

**Figure 4 microorganisms-09-01817-f004:**
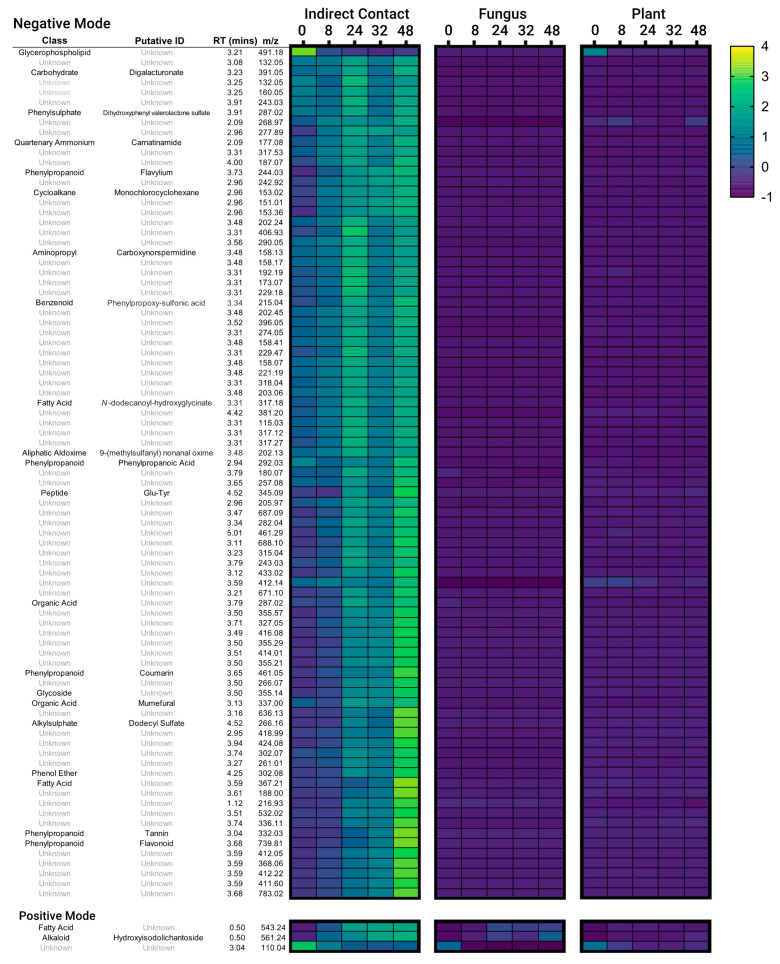
Abundance Z-scores for features identified as significantly elevated in Indirect Contact treatments at 48-h. Z-scores are scaled for each individual feature across all treatments (Indirect Contact, Fungus, Plant) and timepoints (0–48 h).

## Data Availability

Data presented in this study is available within the article or Appendix A. Raw metabolomic data tables are presented in Appendix A.

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
