# Peer review of "Novel Microdialysis Technique Reveals a Dramatic Shift in Metabolite Secretion during the Early Stages of the Interaction between the Ectomycorrhizal Fungus Pisolithus microcarpus and Its Host Eucalyptus grandis"

_microorganisms, 2021, doi:10.3390/microorganisms9091817_

Round 1

Reviewer 1 Report

This paper is interesting, well prepared and described. I have only a few minor comments/questions.

Line 37-38
Authors' wrote: "Trees in temperate and boreal forests worldwide live in symbiosis with ectomycorrhizal (ECM) fungi"

As well as trees in other zones, including tropical forests and savanna. Please look at Corrales et al. 2018 (Ectomycorrhizal associations in the tropics, New Phyt. 220(4):1076-1091) and Steidinger et al. 2019 (Climatic controls of decomposition drive the global biogeography of forest-tree symbioses, Nature 569:404–408).

Line 106
Authors' wrote: "P. microcarpus isolate Si14"

Why you have selected this ECM species? Eucalyptus trees are associated with numerous ECM fungi.

I suppose, the results made using a few ECM fungal species and one tree species, or one ECM fungus on the roots of several tree species, would have allowed for the general conclusions about plant-fungal interactions. One fungal species and one tree species could not be enough.

Line 234
Please, increase the font sizes in Figure 3.

Line 306
Authors' wrote: "While studies on early plant-microbe interactions focus on the impact of microbe elicitors on plant exudation, our results here demonstrate that the impact of the plant on fungal exudation may be just as important."

I think, this is a key conclusion, which should be in Abstract section.

Author Response

We thank the reviewer for their comments. We have addressed their comments as follows:

  • We have expanded our statement in the introduction to encompass all habitats inhabited by ECM fungi, including additional references as suggested
  • Pisolithus is a well characterised ECM fungus and serves as a model species in many publications over the past 20+ years. In addition, both organisms have sequenced genomes and are well adapted to in vitro study. We have now highlighted our reasons for using Pisolithus in the manuscript. The reviewer is correct in the statement that our findings may not extend to all fungal-host combinations, so we have amended our discussion to clarify that these finding are particular to this study system and suggest study in other model ECM systems for the future.
  • Figure 3 was redone with a larger font size.
  • We agree, the large change in fungal exudation is a key conclusion of our paper. Our abstract was re-written slightly to better bring out this important point.

Reviewer 2 Report

This study has clear design, is well performed, have significant results, is written in perfect style and is catchy-reading.

I am not expert in fungal metabolomics, but I believe this part will be commented by somebody who is and the technical execution and presentation of metabolomics features in experiment is well explained and presented.

I have only minor questions and remarks concerning general significance of the research and in vitro cultivations. Sometimes I feel that the general background can be provided by more recent citations, especially in the Introduction.

I was unable to find any typo or grammar mistake – perfect English and clear writing.  

I strongly recommend publishing this research and I congratulate authors for their achievement.

Minor remarks:

Introduction:

Line 39: The interaction of tree and ECM fungus is more complex in light of current knowledge. Is there not a more recent publication than a monograph from 2008 about this topic? Diversity of ECM fungi is driven by their ecological role sometimes divided not only by nutrients in general but also by their specific role in accessibility of elements like P, Mn, etc.

line 46: The expression “tree-host incompatible interactions” sounds too general and conclusive for a fungus species, I think authors mean only occasional incompatibility, please reword, e.g. replace “many” with “occasional tree-host incompatible interactions”

lines 49-67: I do not see any mention about role of  synaptotagmins in early communication of plant-fungus, see https://doi.org/10.3390/plants10020251, https://doi.org/10.3390/jof6030148, https://doi.org/10.3389/fpls.2017.00201

Is there some reason why Pisolithus and Eucalyptus are used as models of early ectomycorrhizal communication?

Material and methods:

Line 92: start sentence with full genus name (“Eucalyptus”)

lines 92-105: I am not familiar with this type of cultivation experiments so my question may not be relevant to this method, but please consider this: how much might endophytic fungi resist surface seed sterilisation and be transferred to seedlings of E. grandis? How were the seedlings prevented to be contaminated by airborne fungal innocula?

Study design: it is a pitty that no control for a non mycorrhizal (pathogenic or saprophytic) fungus was used to sort our features potentially specific for an ECM activity …

Author Response

We thank the reviewer for their comments. We have addressed their comments as follows:

  • Introduction
    1. We have expanded introductory sentence to include the contribution of ECM fungi to trace element uptake and added in some additional recent references on the general role and distribution of ECM fungi
    2. The sentence at line 46 has been rephrased as requested
    3. Thanks for drawing our attention to these references. Synaptotagmins appear to be emerging as important proteins in rhizobial symbiosis and other mutualistic interactions – it would be interesting to investigate their potential role in ectomycorrhizal associations. However, as this section of the paragraph is dealing specifically with secreted metabolites/proteins in ECM associations we have decided not to include these references here.
    4. Pisolithus interactions with Eucalyptus have been well studied for 20+ years, genomes for both organisms are available and they are well adaptable to in vitro study. We have expanded this section to better explain the reasoning behind our choice of study organism and added in some relevant references.
  • Materials and Methods
    1. We have corrected line 92 so it now begins with the full name
    2. There is always a possibility that an endophyte may be present in the E. grandis even after surface sterilisation, however, we routinely grow these seedlings on nutrient rich non-selective media and do not see evidence of hyphal growth, either on plates, or when we are doing microscopy on the tissues. All manipulations of the fungus or plants are done in biological safety cabinets with aseptic techniques to keep the system contamination free. We have included some of these details in the materials and methods.
  • Study design – agreed, a non-colonising fungus would have been interesting in this experiment, but outside of the scope of the current project. If the reviewer is interested in a study of this kind, we would suggest reading Wong et al 2019 Front Ecol Evol, published by some of the current manuscript co-authors, which considers metabolomic responses by fungi of different lifestyles.